# Strategies to Prevent Sarcopenia in the Aging Process: Role of Protein Intake and Exercise

**DOI:** 10.3390/nu14010052

**Published:** 2021-12-23

**Authors:** Patricia S. Rogeri, Rudyard Zanella, Gabriel L. Martins, Matheus D. A. Garcia, Geovana Leite, Rebeca Lugaresi, Sandro O. Gasparini, Giovana A. Sperandio, Luis Henrique B. Ferreira, Tacito P. Souza-Junior, Antonio Herbert Lancha

**Affiliations:** 1Laboratory of Applied Nutrition and Metabolism, School of Physical Education and Sport, University of Sao Paulo, Sao Paulo 05508-030, Brazil; patricia.rogeri@bristolhospice.com (P.S.R.); rudyzanellajr@gmail.com (R.Z.J.); gabrielloureiro.nutri@gmail.com (G.L.M.); matheus_dag@usp.br (M.D.A.G.); geovana.leite@usp.br (G.L.); rebeca.lugaresi@gmail.com (R.L.); sandro.gasp@hotmail.com (S.O.G.); 2Sports Medicine Group, Department of Orthopedics and Traumatology, Santa Casa de Sao Paulo, Sao Paulo 05508-030, Brazil; giovannasperandio@hotmail.com; 3Research Group on Metabolism, Nutrition and Strength Training, Federal University of Parana, Curitiba 81530-000, Brazil; lhboikoferreira@gmail.com (L.H.B.F.); tacitojr@me.com (T.P.S.-J.); 4Laboratory of Clinical Investigation 26-Experimental Surgery, Department of Surgery, Medical School Hospital das Clinicas, University of Sao Paulo, Sao Paulo 05508-030, Brazil

**Keywords:** protein, aging, metabolism, nutrition, exercise

## Abstract

Sarcopenia is one of the main issues associated with the process of aging. Characterized by muscle mass loss, it is triggered by several conditions, including sedentary habits and negative net protein balance. According to World Health Organization, it is expected a 38% increase in older individuals by 2025. Therefore, it is noteworthy to establish recommendations to prevent sarcopenia and several events and comorbidities associated with this health issue condition. In this review, we discuss the role of these factors, prevention strategies, and recommendations, with a focus on protein intake and exercise.

## 1. Introduction

The number of older adults has increased over the last decades. According to the World Health Organization (WHO) [1], until 2025, a 38% increase of individuals over 65 years old is expected, suggesting that a better understanding of that population and strategies to avoid age-related problems to achieve healthy aging are needed.

One of the main problems observed in older adults is related to a relative loss of muscle mass, defined as sarcopenia, which increases risk related to falls, reduces physical capacity, and enhances problems associated with disabilities [2]. Sarcopenia is a multifactorial process associated with several risk factors (i.e., inflammatory cytokines, negative net protein balance, sedentarism, and vitamin D deficiency). Dietary protein intake, insulin resistance, and physical inactivity play a vital role in developing this condition [3].

Over the years, it has been established that an insufficient dietary protein intake is associated with loss of muscle mass in older adults due to lower muscle protein synthesis (MPS) [4]. Therefore, the previous recommendation for protein intake (0.66–0.80 g/kg/day) could be underestimated to sustain the protein net balance across the day, considering problems associated with anabolic resistance [5]. Recent studies suggest that older adults need to ingest 1.0–1.3 g/kg/day of protein to sustain their muscle mass and functionality, indicating that these higher doses represent 40% less muscle mass loss when compared to the lower doses previously recommended [5,6]. Besides the minimum amount of protein intake required to optimize the MPS in older adults, other topics related to optimal doses of protein per/meal have been discussed over the decade. It has been suggested that older adults present the capacity to support and synthesize more protein (>20 g) at each meal [7], supporting the importance of the doses and quality of the protein ingested by older adults.

Moreover, MPS can be stimulated by physical activities, leading to significant increases during aerobic (AT) and resistance training (RT) protocols [8,9]. However, considering associated problems with anabolic resistance of older adults, the increased MPS provided by the RT is insufficient to sustain a positive protein net balance across the day, suggesting that the combination of AT/RT and increases in protein intake may be a better approach to preventing sarcopenia [10].

Thus, the objective of the present review is to analyze optimal nutritional strategies focused on the maintenance of muscle mass in older adults, discussing the protein amounts, dose per meal, and protein quality and source to achieve healthy aging. In addition, the combination of nutritional strategies with training protocols is discussed to provide a better understanding of the interactions between exercise, feeding, and MPS.

## 2. Sarcopenia

Sarcopenia comes from the combination of two Greek words: sarx (flesh) and penia (loss) and was initially described by Evans and Campbell. Later, it was used by Irwin Rosenberg, who defined this condition as age-related muscle wasting. Sarcopenia is currently recognized as a critical geriatric problem and an important condition to predict frailty in the elderly [3] and minimize the impact on physical activity [11]. In addition, sarcopenia is associated with increased mortality risk in frail older adults.

On 18 November 2009, researchers and geriatricians defined sarcopenia as “the loss of skeletal muscle function and mass associated with age”. It is a complex syndrome associated with muscle mass loss alone or in conjunction with increased adipose tissue. Its causes are multifactorial and include disuse, altered endocrine function, chronic diseases, inflammation, insulin resistance, and nutritional deficiencies. Although cachexia can be a component of sarcopenia, these conditions are distinct [3].

The European consensus on sarcopenia recommends the inclusion of the factors that cause the decrease in skeletal muscle mass and consequent decrease in its function (strength or performance) for the diagnosis of this syndrome. The justification for using the two factors mentioned above is that muscle strength is not dependent on muscle volume, and that strength and muscle mass ratio is not linear. Therefore, defining sarcopenia only by decreasing muscle mass could have limited clinical value [2].

Like any other complex syndrome, sarcopenia's onset mechanisms and progression are varied and related to the (in)balance of protein synthesis and degradation, neuromuscular integrity, and muscle fat content. Such progression is due to factors such as age advancement, inadequate nutrition, disuse, and endocrine dysfunction [2]. Although the incidence of sarcopenia is higher in the elderly, adults can also develop this syndrome, which can be associated with other diseases such as osteoporosis. Thus, sarcopenia can be considered primary if related to advanced age and with no other evident cause or associated disease. In contrast, secondary sarcopenia can be linked to three other factors: lack of motor activity (long periods in bed, sedentary lifestyle, and others), disease, or inadequate nutrition, caused by inadequate energy and/or protein intake, malabsorption of nutrients, gastrointestinal diseases, and the use of drugs that produce anorectic side effects [12].

The sedentary lifestyle related to the aging process tends to damage mitochondrial function and insulin resistance. This situation is almost always present in metabolic diseases, sarcopenia, and obesity, resulting in increased risk of mortality in the elderly [13]. Currently, the COVID-19 pandemic has led thousands of people worldwide to measures of distancing and social isolation implemented by governments. As a result of these actions, there is a reduction in physical activity, interruption of routine eating habits, stress, and altered sleep patterns, which offer an environment conducive to sarcopenia installation [14]. Like any other complex syndrome, sarcopenia’s appearance and progression depend on various mechanisms related to the balance of protein synthesis and degradation, neuromuscular integrity, and muscle fat content. Such mechanisms include age progression, inadequate nutrition, disuse, and endocrine dysfunction [12]. The lower levels of anabolic hormones (such as testosterone, insulin-like growth factor (IGF-1), and estrogens) that regulate MPS are noticed throughout the aging process [15].

Sarcopenia is associated with changes in skeletal muscle physiology and cellular mechanisms. These changes can be observed at the metabolic, cellular, vascular, and inflammatory levels [16]. Metabolic alterations in the anabolic pathway (mTOR), an important regulator and signaler of muscle cell growth, can often impair the sarcopenic muscle [17,18,19]. The concomitant loss and atrophy of muscle fibers, specifically the loss of type II fibers, is classically one of the most evident signs of sarcopenia [16]. Myofibrillar protein synthesis is also impaired by the inability of satellite cells to react positively to growth factors and cytokines (myokines), which are essential to stimulate the production of these contractile proteins [17,19].

IGF-1 is a classic pathway for skeletal muscle anabolism that also suffers different adjustments in the sarcopenic muscle. Through animal models that suppressed the expression of insulin receptor substrate (IRS-1), the results showed increased longevity in animals that had this anabolic pathway suppressed. The current explanation for this phenomenon is that reductions in IGF-1 perceived with aging may be an attempt to increase the Silent Information Regulator 1 (SIRT-1) proteins. These proteins are essential for cell survival, especially by combating catabolic cytokines (e.g., tumor necrosis factor-alpha (TNFα)) that are constantly elevated in the circulation and skeletal muscle in the aging process. Therefore, in addition to playing an essential role in myoblast survival, SIRT-1 can also participate in the adequate differentiation, hypertrophy, and atrophy arrest in in vivo processes during stressful stimuli contained in chronic inflammation or disuse [20].

Older men also have decreased testosterone, which is one of the critical factors for decreasing muscle mass in this population. Long-term hormone replacement therapies increase muscle mass and muscle strength in the elderly. Testosterone regulates protein synthesis via the androgen receptor, and its administration has been used to increase this anabolic signaling in response to strength training in older adults. However, testosterone administration in those individuals may be unable to entirely reduce muscle wasting if not combined with nutritional strategies and strength training [21,22,23].

Another condition that is harmful to the sarcopenic muscle is the infiltration of fat (in and/or between) the muscle fibers. Changes in the differentiation of satellite cells into adipocytes explain this phenomenon’s pathophysiology. Adipocyte infiltration, known as lipotoxicity, will promote the release of toxic adipokines that affect muscle cell function [24].

Sarcopenia’s vascular mechanism is explained by a capillary density reduction, associated with low blood perfusion in the musculature, increased oxidative stress, and mitochondrial dysfunction. These modifications are associated with reduced peroxisome proliferator-activated receptor-gamma 1-alpha (PGC1α) coactivator expression, which is involved in type I muscle fibers formation. Together, they participate in mitochondrial activation genes and fatty acid-binding proteins muscle factors involved in the use of fatty acids for energy production in the mitochondria [17]. Finally, sarcopenia is associated with inflammation demonstrated by increased serum C-reactive protein (CRP) levels, interleukin-6 (IL-6), and TNFα. Elevated levels of these inflammatory markers have been associated with reduced muscle mass and strength in rodents [16]. In addition, in vitro studies have shown that TNFα inhibits myogenesis and increases nuclear factor kappa beta (NF-κβ), which is essential in skeletal muscle atrophy [25,26].

Anabolic resistance is also part of the sarcopenia process. It is a diminished response to the stimulating effects of protein synthesis from strength exercise and protein intake in the elderly population [14]. For example, it has been reported that older adults compared to young adults (mean 71 years vs. 22 years) require twice the need for protein intake (0.60 vs. 0.25 g/kg/weight) to stimulate MPS. An essential factor to be considered in anabolic resistance is the reduction of skeletal muscle capillarization, an issue that can hinder the hypertrophic effect of strength training. This issue was demonstrated by researchers who submitted older adults to participate in a strength training program for 12 weeks. At the end of the program, the elderly with reduced muscle capillarization did not present the same level of hypertrophy as those elderly with increased capillarization [27].

## 3. Muscle Protein Metabolism and Anabolic Resistance of Aging

Skeletal muscle mass is regulated by a tightly and dynamic process involving MPS and muscle protein breakdown (MPB). Skeletal muscle proteins are continuously being turned over since MPS and MPB simultaneously occur throughout the day. When the rate of MPS exceeds MPB, a positive net muscle protein balance occurs, meaning that new proteins are being incorporated into muscle tissue, resulting in muscle hypertrophy in the long term. On the other hand, when the rate of MPB exceeds MPS, a negative net muscle protein balance occurs with a loss of muscle proteins, which in the long term may induce muscle atrophy. Finally, when there is a balance between MPS and MPB throughout the day, there is a neutral muscle protein balance and the maintenance of skeletal muscle mass in the long term [28]. In this sense, during postabsorptive conditions, the rate of MPB generally exceeds those of MPS, resulting in periods of net muscle loss. However, after consuming a meal containing proteins, an MPS increase and MPB suppression generate a positive protein balance. Thus, in healthy and young individuals who consume sufficient daily amounts of protein (around 0.8 g/kg/body weight), the fluctuations between periods of negative and positive net muscle protein balance (which are the results of postabsorptive and postprandial periods, respectively) are generally equivalent. Therefore, the skeletal muscle mass remains stable [29].

Protein ingestion and resistance exercise are the two most potent anabolic factors capable of stimulating MPS and promoting positive muscle protein balance [8]. However, older individuals seem resistant to these external stimuli [30]. Data regarding the effects of resistance exercise [31] and protein ingestion [32] suggest that older people are less sensitive to the anabolic effects of exercise and protein/essential amino acids (EAAs) ingestion. This condition, called “anabolic resistance” of aging, is characterized by a blunted stimulation of MPS to protein ingestion and resistance exercise [10]. The reduced effects of protein consumption on MPS may contribute to a chronic state of negative muscle protein balance, with rates of MPB being constantly higher than MPS, which has been pointed out as a primary contributor to muscle loss in aging [10]. This condition was clearly demonstrated in a study conducted by Wall et al. (2015) [30], which pooled together data from six studies performed in the same laboratory with similar designs. This analysis found a significant reduction in the synthetic muscle protein response to ingesting 20 g of casein protein compared with young counterparts. Importantly, although there is a reduction in the sensitivity of skeletal muscle to nutrient ingestion in older people, basal MPS seems to be similar both in the young and older populations [30,33]. Moreover, skeletal muscle protein metabolism analysis revealed that the rate of MPB, both at rest and after a resistance exercise session, seems to be comparable between the young and elderly [34]. Therefore, the reduction of MPS to external anabolic factors characterizes the anabolic resistance of aging and may be partly responsible for the progression of sarcopenia in the latter stages of life.

The possible contributors to the onset of anabolic resistance are still unclear. However, factors such as reduced capillarity and/or vasodilation capacity, attenuated muscle uptake, reduced ribosomal protein content or activity, chronic low-grade inflammation, obesity, and physical inactivity may play a role in the decreased muscle sensitivity to protein and amino acids ingestion [35]. This section highlights the roles of physical inactivity, inflammation, and obesity as aggravating anabolic resistance factors, which can be modifiable through lifestyle changes. We also briefly discuss the possible role of gut microbiota in the genesis of anabolic resistance.

### 3.1. Decreased Muscle Contraction in Aging (Physically Inactive X Sedentary Behavior)

Currently, the classification of an individual as “sedentary” or “physically inactive” occurs through different criteria, which aim to quantify the contractile muscle stimulus (weekly and/or daily) of the subject observed. People considered “sedentary” are characterized by having sedentary behavior for the most part of the day (such as watching television, working sitting in the offices and/or stores, or spending extended periods lying down during the day), thus presenting a low metabolic demand rate in the muscle tissue (≤1.5 metabolic equivalents) [36,37]. In addition, the term “physically inactive” is used for subjects who perform an insufficient amount of exercise compared to current health recommendations (<150 min of moderate or vigorous physical activity per week, through aerobic exercise, training strength, team sports, and other activities) [36,37]. Considering these terminologies, the effects of reducing muscular metabolic demand (physical inactivity or sedentary lifestyle way) must be interpreted with full attention and together. Notably, a subject can be considered “physically active” but “highly sedentary” as well as “physically inactive” and with almost non-existent sedentary behavior. For each case, associations between these different “lifestyles” (through the motor activities performed) and distinct anthropometric and/or metabolic outcomes are pointed out [38].

In this context, a large study based on a database of 1.9 million individuals pointed out that physical inactivity has a high prevalence in the adult population worldwide (27.5%), with wide variation in its incidence between different countries analyzed (10 to >50% of the population) [39]. In addition, recent (robust and well-defined) studies indicate that the degree of physical activity observed among older adults (≥60 years old) is reduced during the aging process, with a parallel increase in sedentary behavior [40,41,42]. This scenario raises a “state of alert” among researchers worldwide, who aim to understand how the global adult population is aging and the outcomes of a reduced muscle metabolic demand on the quality of life and muscle area/function of older adults.

Globally, the relationship between physical inactivity and the significant incidence of chronic diseases is already well established [43]. On the other hand, there is a 6–10% incidence reduction of diseases such as type 2 diabetes (DM2) and some types of cancer, which occurs by increasing physical exercise in the world population [44]. In addition, well-controlled clinical trials indicate that the practice of lifelong physical training (among former highly active athletes, for example) can preserve muscle tissue’s function and volume in aging when compared to the effects of physical inactivity on muscle tissue of healthy elderly (absence of chronic diseases) and/or young adults [45,46]. Regarding sedentary lifestyle, Seguin et al. [47] pointed out that older women with high levels of sedentary behavior (8–11 h·day^−1^) have lower muscle functionality when compared to older women of the same age group who presented reduced sedentary behavior in their waking period (<6 h·day^−1^). Complementarily, Gianoudis et al. [48] indicated that a 60-min increase in the daily sedentary behavior of the elderly (variation of 6–10 h·day^−1^) was related to a 33% greater risk of presenting a reduction in muscle volume and muscle strength during aging (sarcopenia). These results suggest that aging is not necessarily accompanied by sarcopenia, reinforcing the notion that the absence of exercise and/or sedentary behavior are “key elements” for the progression of sarcopenia/atrophy of the aging muscle tissue.

Regardless of the causes behind the reduction in muscle activity in aging, it is commonly accepted that older individuals have higher rates of both “muscle volume” and “muscle functionality” losses in periods of physical inactivity/disuse when compared to younger individuals [49,50]. Moreover, studies indicate that older subjects present lower muscle volume/functionality recovery (after a period of disuse) than young adults [50,51,52]. Recent findings indicate that physical inactivity/disuse intensifies the muscular anabolic resistance state of the elderly, possibly through the increase of pro-inflammatory markers (TNFα and CRP) and through the reduction of postprandial insulin sensitivity [53,54]. Noteworthily, the current literature indicates an interesting relationship between the increase in pro-inflammatory markers (or the dysregulation between pro- and anti-inflammatory markers) and the accentuated loss of muscle volume/function in the elderly (for more details, see topic “Inflammation, aging, and muscle tissue”). In randomized trials focusing on measuring MPS rates and lean mass gain, this crosstalk between muscle, aging, and “inflammatory balance” needs further investigation.

Aiming to elucidate the impact of different periods (or “models”) of reduced contractile activity in aging, several studies investigated the changes in muscle metabolism under three circumstances, which are typical throughout the aging process: the number of daily steps [29,51,54,55,56], specific muscle groups immobilization [52,57,58,59,60], and/or “bed rest” periods [61,62,63,64,65,66]. These three different models of reduced muscle contractile activity led to the so-called “catabolic crises”: events that favor the transition of older adults to the sarcopenic state, through brief moments of reduction in muscle activity with high catabolism of muscle mass, often difficult to establish “full recovery” for older individuals [67]. Based on this concept, older adults may have one or more of these catabolic crises “triggering” events as the decades go by (by hospital admissions, limb immobilization, and momentary difficulty in locomotion), increasing their risk for muscle atrophy in older-aged subjects.

Among the studies that analyzed the impact of reducing daily steps among participants aged 65–73 years [51,54], it was pointed out that individuals who perform a small number of daily steps (<1000–1500 daily steps, for 14 days) show a ~14–26% MPS decay when compared to subjects who practice ≥6000 steps daily [51,54]. Limb immobilization studies showed that quadriceps immobilization for 14 days led to a 5% decrease in thigh muscle volume among older adults. Other studies showed losses of around 2% in the first five days of immobilization/disuse, combined with an initial loss of muscle strength in the order of 8.3% [60]. On the other hand, bed rest muscle contractile activity reduction models (a more rigorous model of sedentary lifestyle and physical inactivity) showed that only five days of bed rest provides a 4% reduction in the thigh lean mass with a more accentuated reduction in the MyHC I fiber type (−26.3% ± 17.2%) [68]. Relevant losses in muscle functionality are well documented in the elderly during the first ten days of bed rest [63,64,65].

From these initial data, individualized strategies (nutrition, low-intensity exercises, and reduction of sedentary behavior) for the elderly population aiming to recover muscle functionality and/or protection from muscle catabolism should be better investigated during these periods of physical inactivity/disuse. Nutritional strategies might include the appropriate consumption of protein or amino acids. Physical exercise practice should focus on improving specific older people’s limitations/needs. In addition, sedentary behavior seems to be potentially relevant in the context of aging and musculoskeletal changes. In this case, studies indicate that sedentary behavior could be reversed by increasing the number of steps per day (>6000 steps) or engaging in fewer than 6 h·day^−1^ of sedentary behavior during the individual’s waking time. In the context of aging, this should be better explored in future clinical trials.

### 3.2. Inflammation, Aging, and Muscle Tissue

Aging is characterized by physiological changes that induce low-grade chronic inflammation through the constant presence of pro-inflammatory factors/agents. The main changes inherent to aging occur both because of changes in cellular aging/cell renewal processes (senescence process), as well as significant changes in lipid metabolism and/or storage in the muscle tissue [69,70,71]. Notably, these changes induce more significant chronic signaling in the activity of immune cells (infiltrating lymphocytes and macrophages), in addition to a higher production of reactive oxygen species (ROS) and cell damage, providing systemic increases in pro-inflammatory markers throughout the aging process (such as NF-κB, Interleukin-1 [IL-1], IL-6, interleukin-8 [IL-8], CRP, and TNFα) [70,72,73,74].

Higher levels of some of these pro-inflammatory markers are related to unfavorable muscle metabolism and function changes during the aging process (sarcopenia) [75,76,77,78,79,80]. In these cases, some studies showed a higher concentration of inflammatory markers (CRP, IL-6, and TNFα) in the elderly who have a more significant loss of strength/muscle mass, with a simultaneous decrease in their physical capacity [80,81].

In a recent study, Sciorati et al. (2020) [82] clarified the physiological impacts of TNFα production on muscle fibers by monitoring the aging of healthy mice from 12 to 28 months of life (corresponding to 40–90 years of age in humans) under two conditions. In the study, control mice (C57BL/6) showed functional impairment of muscle tissue with parallel muscle atrophy (±20% drop) [82]. In contrast, the rodents that were treated weekly with a pharmacological blocker for TNFα (Etanercept) did not present atrophy and loss of muscle fibers (mainly those of type IIB and IIA), which provided improvements in the muscular function of the mice [82].

Another component correlated to increased pro-inflammatory markers (CRP, IL-6, and TNFα) and the loss of functionality in the elderly’s muscle tissue is myosteatosis. It is defined as changes in fat infiltration, leading to increasing intramyocellular and inter-muscular fat [83,84]. Delmonico et al. (2009) [85] studied changes related to thigh muscles composition, strength, and muscle quality of 1678 elderly (mean age of 73.5 years), pointing to a direct relationship of gains and/or maintenance of body weight with the increase in subcutaneous fat (common condition to sarcopenia). Regarding the fat infiltration in the muscle tissue, an increase in intramuscular fat (16.8–74.6%) and loss of maximum torque strength (13.4–16.1%) were observed regardless of the reduction, maintenance, or increase of individuals’ subcutaneous fat [85]. In addition to these findings, Gueugneau et al. (2015) [86] compared the lipid infiltration into the muscle fibers of 5 young people, 15 healthy elderly, and nine elderly subjects with metabolic syndrome (MS), pointing out higher intramyocellular lipid contents in the elderly (both healthy and with MS). In addition, a more pronounced increase in the atrophy of IIX type and IIA-IIX type muscle fibers was observed among the group of older people with MS compared to the group of healthy elderly subjects [86].

Since both the lipids storage and the plasma level of some pro-inflammatory markers are related to the functionality and muscle volume of the elderly, researchers and health professionals should look more closely this population. During the past two and a half decades, factors such as physical inactivity, insulin resistance, increased plasma levels of IL-6, increased intramuscular fat storage, and adequacy of protein consumption have been identified as “therapeutic targets” in combating sarcopenia during the aging process [87,88].

Current research shows that greater “body adiposity” [89,90,91,92], “scenarios of insulin resistance”, and “sarcopenia” are associated with increased systemic levels of various inflammatory markers (particularly IL-6 and TNFα). These markers, in theory, would be able to stimulate metabolic pathways related to muscle atrophy and reduced muscle fiber regeneration [93,94]. As a result, greater adiposity, MS, DM2, and other chronic diseases associated with aging, promote an even more robust increase in pro-inflammatory markers, which are potentially harmful to protein synthesis and muscle mass gain/maintenance.

From recent discoveries about the imbalance of pro- and anti-inflammatory mediators and their possible impacts on the synthesis and/or maintenance of muscle tissue, contemporary research has studied the effects of dietary training on seniors’ inflammatory markers and muscle tissue. Notably, the consumption of low amounts of protein is related to the increase in pro-inflammatory markers (CRP, IL-6, and TNFα). Therefore, specific adjustments in protein consumption/fractionation are indicated for the elderly population [75,95]. In addition, it is pointed out that muscle contraction induced by physical training can increase the production of a series of anti-inflammatory markers (IL-6, Interleukin-10 [IL-10], and transforming growth factor beta [TGF-β], for example), which are favorable for adaptive processes of the muscle tissue [96,97]. There are also encouraging results pointing to a reduction in some pro-inflammatory markers (TNFα, CRP, IL-1, IL-6, and IL-8) in the elderly population, either through the acute realization of specific training protocols or by greater physical conditioning [54,96,97,98,99,100,101]. For a complete review of exercise protocols (aerobics, and combined and strength exercises), refer to Bruunsgaard, 2005 [99]. For more information about the impact of physical inactivity/sedentary behavior on muscle tissue, the reader should be directed to the section “Effect of physical inactivity/sedentary behavior on muscle tissue in aging”.

In summary, regular physical exercises and adequate dietary protein intake must be essential points of attention in geriatrics studies. In this regard, the current academic literature points out that physical training and protein support are factors of high importance both for the balance of pro- and anti-inflammatory markers (such as the TNFα/IL-10 ratio or the IL-6/IL-8 ratio) and for the synthesis/degradation stimuli balance in the muscle mass of the elderly [97,101,102,103,104].

### 3.3. Digestion, Absorption, and Gut Microbiota

In recent years, knowledge related to gut microbiota and its importance in the host function has been increasing. It is known that gut microbiota suffers many alterations during a person’s life; with aging, an inversion of phyla predominance occurs. For example, proteobacteria phylum (pro-inflammatory bacteria genus) increases overlapping with bacteroid and firmicute phylum (the predominant phylum in a healthy adult), associated with decreasing in health-promoting bacteria (e.g., Bifidobacterium, Faecal-bacterium genera) and short-chain fat acids (SCFA) production. This dysbiosis is associated with increased gut permeability in the elderly [105]. Considering the strong interaction between the gut microbiota and the immune system, studies suggest that with the aging process, gut-microbiota inflammation precedes the low-grade systemic inflammation [105].

There is also a well-known connection between the gut and the muscle tissue, which connects the aging-dysbiosis and sarcopenia process and is related to anabolic resistance. Dysbiosis generates low-grade systemic inflammation (from the increased circulating endotoxins) that culminates in an increase in ROS production, expression of NFKB, inhibition of anabolism, reduction of protein synthesis, and facilitation of insulin resistance [106]. Accordingly, therapeutic targets related to gut microbiota (e.g., probiotics, fermented foods, transplantation) have been appointed as helpful to slow the aging process.

According to Gorissen et al. (2020) [107], protein absorption kinetics is attenuated in the elderly compared to young individuals since older adults have reduced pathways related to carbohydrate metabolism and amino acid synthesis by gut microbiota [108]. Due to the critical relationship between protein adsorption kinetics and anabolic stimulus, probiotics may act on protein absorption kinetics. A study [109] showed that two weeks of probiotic use (5 billion CFU L. paracasei LP-DG^®^ (CNCM I-1572) plus 5 billion CFU L. paracasei LPCifS01 (DSM 26760)) was able to increase methionine, histidine, valine, leucine, isoleucine, tyrosine, total BCAA, and total EAA maximum concentration (Cmax) and area under the curve (AUC) after the consumption of 20 g of pea protein. However, it is important to note that the sample of this study was young and physically active men. Thus, it is possible that in elderly subjects, the results could be different. Additionally, it is unknown if similar results would be found after consuming a higher-quality protein source (meat or milk proteins).

A powerful enterocyte energy source and anti-inflammatory compound is butyrate. Therefore, declines in butyrogenic bacterial species, a characteristic of the aging process, may be viewed as a pivotal contributor to age-related anabolic resistance. Besides, some specific amino acids appear to be dependent on microbial sources (lysine, leucine). For example, aging is associated with a decrease in Prevotella, a microorganism involved in lysine biosynthesis and related to leucine. The gut microbial (Prevotella, Allistiples, and Barnesiella) synthesizes up to 15% of leucine (the most critical amino acid to signal the intracellular anabolic signal). Another point to be considered is the splanchnic amino acids extraction; in older individuals, we can see an increase in leucine oxidation in the gut and/or liver. Studies suggest a heightened rate of leucine oxidation and thus more significant splanchnic sequestration of amino acids associated with aging dysbiosis [110].

Dysregulation of the IGF-1 anabolic signaling cascade and resultant sarcopenia may be caused by dysbiosis of the gut microbiome and depletion of IGF-1-related microbes such as lactobacilli, possibly the *L. Plantarum* [111]. According to a review by Badal et al. (2020) [108], a greater abundance of *L. Plantarum* is seen in healthy, long-lived individuals [108]. A study conducted by Hopkins et al. (2001) [112] with healthy elderly subjects and geriatric patients demonstrated a reduced number of Bifidobacterial (other important acid-lactic bacteria genera) [112]. Species diversity was markedly lower in the clostridium difficile associated diarrhea group, characterized by high numbers of facultative anaerobes and low levels of bifidobacteria and bacteroides. According to the authors, the reductions in these organisms may be related to increased disease risk in older adults [112].

Preservation of these good microbial strains with diet strategies, polyphenols, probiotics, prebiotics, and fermented foods should be considered to reduce anabolic resistance in aging [113,114,115].

In addition, studies indicate that physical activity is a viable therapy to counteract age-related gut dysbiosis, improving the bioavailability of nutrients through the action of specific intestinal bacteria (e.g., polyphenols). Additionally, exercise can induce favorable changes in gut microbiota composition (increasing health-promoting bacteria) and metabolite production (increasing SCFA producing taxa and microbial production) [113,114,115].

## 4. Daily Protein Requirements

The current recommendation for protein consumption is based on the lowest amount of protein required to maintain a neutral nitrogen balance. Therefore, the amount of protein ingested should be enough to balance the loss of nitrogen throughout the day and enough to maintain bodily proteins. The actual protein consumption recommendations are based on a meta-analysis conducted by Rand et al. (2003) [116], who evaluated 19 nitrogen balance studies performed in adults. The results of this study originated the current recommendations of 0.66 g/kg/d and 0.8 g/kg/d as the estimated average requirement (EAR) and recommended dietary allowance (RDA), respectively. However, some criticism has been raised in recent years whether these recommendations are really optimal for older adults. Since only one of the 19 included studies evaluated older adults, and the nitrogen balance technique may present some limitations [117], researchers have been raising the question of whether there is, in fact, an ideal level of protein intake that is higher than the current RDA [118]. One argument often cited against the actual recommendation based solely on nitrogen balance is that it does not consider any other health parameters, such as muscle function. Additionally, it represents the minimum amount of protein to balance nitrogen losses and not the optimal amount to promote the synthesis of new bodily proteins.

Given this scenario, newer techniques have been used to re-evaluate protein recommendations to the elderly. To this end, some studies have been conducted utilizing the indicator amino acid oxidation (IAAO) as a novel approach to establish protein requirements [119,120,121]. Studies applying these newer techniques point to a greater protein requirement in the older population. For example, Rafii et al. (2015) [120] recruited women aged 65 or older and observed an increased protein requirement of 1.29 g/kg/d. Similarly, when evaluating older men, the researchers found a requirement of 1.24 g/kg/d [119]. Although a small number of studies were conducted in older adults that sought to evaluate protein needs utilizing the IAAO technique, there were divergent results when compared to the current recommendations based on nitrogen balance studies. This highlights the need for more studies applying this method to be conducted in a more significant number of individuals to confirm, or not, these results and possibly contribute to updating the actual protein recommendations for the elderly.

The limitations of the nitrogen balance technique, and the new findings from the IAAO studies, combined with the data showing lower sensitivity of the skeletal muscle of older people to the stimulatory effects of protein consumption on MPS [57], corroborate the idea of an increased protein requirement in aging. In this sense, new researchers’ recommendations engaged in the study of nutrition and healthy aging point to increased daily consumption of dietary protein, reaching intakes between 1.2 and 2.0 g/kg/d depending on the health status of older adults [117].

From an observational standpoint, evidence points to a beneficial role of higher protein intake than the current RDA to preserve muscle mass in aging. Cross-sectional data showed that a protein intake higher than 0.8 g/kg/d was positively associated with higher lean body mass compared to lower intake levels [122]. The prospective Health, Aging, and Body Composition Study (Health ABC) [6] followed a group of 2066 older individuals for three years. It was found that, after this period, the participants who consumed the highest amount of dietary protein (1.1 g/kg/d) presented lower lean body mass (approximately 40%) than the group with the lowest amount of protein intake (0.7 g/kg/d). In the Framingham Offspring Study (FOS), the authors sought to evaluate the independent association of dietary protein intake on long-term changes in physical functioning over more than a decade in middle-aged and older adults [123]. Seven physical tasks related to strength and muscular endurance were selected. After 12 years of follow-up, an association was observed between poor protein intake and higher prevalence of disabilities, mainly in heavy work at home and to walk 0.5 miles. In addition, the individuals consuming higher amounts of protein were 40% less likely to be physically dependent in one or more functional tasks and 50% less likely to be dependent in two or more functional tasks, over 12 years [123].

Interestingly, however, a more recent analysis by the same study group (Health ABC) reported no association between total protein consumption and type of protein and changes in thigh cross-sectional area in the elderly after five years of follow-up [124]. Nonetheless, some design limitations may have influenced the results. For example, there was only one dietary assessment over the five years, and the muscle analysis was conducted with computerized tomography that analyzed only a small portion of the muscle [124].

Randomized clinical trials (RCTs) had also been conducted to investigate the effects of increased protein consumption (both in the form of whole foods and supplements) on measures of skeletal muscle mass (or lean body mass) in the elderly. In this context, many trials reported the beneficial effects of increased protein consumption to improve lean body mass retention in older adults. Tieland and colleagues [125] supplemented frail older individuals with 15 g of milk protein twice per day and evaluated lean body mass and physical function outcomes [125]. After 24 weeks, although no effects on lean body mass were observed, there was an improvement in physical performance. Posteriorly, Mitchel et al. (2017) [126] investigated the effects of protein consumption at the current RDA (0.8 g/kg/d) or twice the RDA (2RDA—1.6 g/kg/d) on measures of skeletal muscle mass and physical function in the older men [126]. To this end, 29 men with a mean age of 70 years were provided with all foods for 10 weeks to ensure adequate protein intake. The results showed that both groups ended up having a modest energy deficit of around 150–200 kcal/d. Moreover, the RDA group lost around 600 g of appendicular lean mass, while the 2RDA group showed no differences. These findings suggest that doubling the protein intake did not increase appendicular lean mass, but it did improve its retention over time [126]. More recently, Ten Haaf et al. (2019) [127] assessed the effects of 12 weeks of daily protein supplementation on lean body mass, strength, and physical performance in 114 older adults training for walking events of 30, 40, or 50 km/d [127]. Supplementation protocol consisted of 31 g of milk protein divided into two daily doses, one during breakfast and the other within 30 min after training. As a result, the supplemented group increased total daily protein intake to 1.3 g/kg/d, while the placebo group remained at 0.9 g/kg/d. Regarding lean body mass, supplemented group increased by 0.54 kg (0.93%) and placebo group by 0.31 kg (0.44%). Additionally, the supplemented group lost more body fat than the placebo group [127].

Collectively, these data suggest that increased protein intake may confer benefits for older adults regarding the maintenance of lean body mass and physical function. However, recent and more long-term studies have contained contrasting findings. For example, a 6-month RCT recruited 92 functionally limited men 65 years or older and compared the intakes of 0.8 and 1.3 g/kg/d of protein on markers of physical performance and lean body mass [128]. The increase in protein consumption in the high-protein-diet group was achieved by consuming 0.5 g/kg of a casein and whey protein blend supplement. As a result, after six months, there was no effect of increased protein intake on lean body mass, strength, and physical performance, although consuming a high protein diet results in higher body fat loss than a standard protein diet. More recently, Mertz and colleagues [129] conducted a one-year randomized controlled trial investigating the effects of protein supplementation on muscle size and strength in community-dwelling adults aged 65 years and older. Participants were randomized to consume either 2 × 20 g of whey protein or 2 × 20 g of maltodextrin per day, together with breakfast and lunch. The primary outcome was the changes in quadriceps cross-sectional area (qCSA) measured by magnetic resonance imaging (MRI) scans, although strength, functional performance, and body composition were also measured. After one-year, total daily protein intake increased from 1.1 to 1.5 g/kg/d with the consumption of whey protein. However, despite increasing protein intake, no differences were observed between the groups in any measurement performed [129].

Additionally, two meta-analyses corroborate the findings of these two studies, suggesting that protein supplementation alone may not be enough to promote positive changes in skeletal muscle mass in older adults [130,131]. However, some limitations regarding the studies included in these meta-analyses may have influenced the results. Factors such as the amount and source of dietary protein/amino acids ingested can directly influence the anabolic potential of the supplement, which may compromise long-term improvements in skeletal muscle mass and strength [132]. As previously mentioned, older individuals suffer from anabolic resistance, which impairs the rise in MPS when low doses of protein/amino acids are ingested. Considering that several included studies provided low doses of protein/amino acids, we must acknowledge that protein was provided in suboptimal doses to stimulate MPS effectively.

Finally, we must remember that similarly to young individuals, resistance exercise is a necessary stimulus to induce increases in skeletal muscle mass since it sensitizes the muscle to incorporate the ingested amino acids into newly contractile proteins more efficiently [133]. This scenario is characterized by a positive muscle protein balance that, chronically, will result in skeletal muscle hypertrophy [29]. In this sense, increases in protein consumption without the concomitant stimulus of exercise (mainly RT) may be important to minimize the losses of this tissue in the long term through the maintenance of a neutral muscle protein balance [29], which may reduce the likelihood that older adults will develop sarcopenia.

Despite some inconsistencies in the literature regarding the impact of increased protein consumption and markers of skeletal muscle health in the aging population, as stated by Wolfe and colleagues [134]: “We believe that the overall conclusion from these various studies is that there is an optimal level of protein intake that is greater than that of the RDA. Importantly, to our knowledge, there has never been a study in which the RDA for protein intake was compared with a higher level of protein intake, and the RDA was found to be superior in terms of any endpoint” [134].

Thus, based on physiological studies showing skeletal muscle decreased sensitivity to low doses of protein/amino acids in older individuals, associated with newer analysis techniques, and some observational and clinical data pointing to possible beneficial impacts of higher protein intake, we suggest that older adults increase their total daily protein intake to levels above the RDA.

## 5. Maximizing Anabolic Effects of Protein Ingestion through an Optimal Consumption Pattern

Beyond total daily protein intake, several studies have been conducted in the last few years to determine if an “optimal pattern” of protein ingestion can maximize its anabolic properties. Thus, in this topic, we will discuss whether such a pattern exists, the role of source and quality of protein, and the combined effects of higher protein intakes and RT.

### 5.1. Protein Dose to Optimal Stimulation of Muscle Protein Synthesis (MPS)

In the last decade, many trials have sought to evaluate whether there is an optimal amount of protein to be ingested in a single meal to effectively, and maximally, stimulate MPS. Dose-response studies in both young [7,135] and older adults were conducted and have found interesting results. While younger individuals present a saturable dose of protein that maximally stimulates MPS around 20 g [7,135], older people seem to need higher amounts [136,137]. Specifically, Pennings et al. 2012 [137] randomly assigned healthy older men to ingest 10, 20, or 35 g of whey protein at rest and measured muscle protein accretion over the next 4 h [137]. As a result, the researchers found that, contrary to young adults, older individuals do not saturate the MPS response after ingesting 20 g of high-quality protein. In fact, there was a greater muscle protein accretion following the ingestion of 35 g of whey protein. Corroborating these findings, Moore et al. (2015) [138] compiled data from six studies that evaluated postprandial MPS in young and older individuals to provide a more accurate and individualized per meal protein recommendation per body weight to stimulate MPS optimally [138]. This combined data revealed that younger individuals require around 0.24 g/kg/meal of protein, whereas older adults require approximately 0.4 g/kg/meal. This data is consistent with the findings from Wall et al. (2015) [30] and corroborates the thesis that older people display reduced sensitivity to lower doses of dietary protein. Although there is a higher per-meal protein requirement in the elderly, the absolute maximal postprandial stimulation of MPS showed similar values between young and older individuals [138]. This information suggests that older people retain the capacity to properly stimulate MPS despite a higher protein dose once optimal protein quantities are provided.

The rationale behind MPS stimulation relies on increases in intracellular concentrations of EAA [139], especially leucine [140]. Data suggests that leucine acts as a “trigger” to initiate the intracellular molecular anabolic cascade that ultimately increases the rates of MPS [141]. This hypothesis points to an increase in MPS proportional to the intracellular leucine concentrations provided that all other EAA are available. Thus, to properly stimulate MPS, a threshold of leucinemia needs to be reached. Recent data utilizing IAAO show that older adults require two times more leucine throughout the day than their younger counterparts [142]. Moreover, some clinical and lifestyle conditions such as DM2 [143], obesity [55], physical inactivity, and bed rest [133] seem to decrease the sensitivity of the skeletal muscle to stimulatory effects of EAA on MPS even more. Therefore, even higher doses of EAA/leucine may be needed to provide an optimal anabolic stimulation of MPS.

Another critical aspect that needs to be addressed is that since older people need protein/leucine in higher amounts, providing higher doses than those reported by Moore et al. (2015) [138] would be even better to stimulate MPS. However, this hypothesis was already discarded by studies showing that once the required concentration of leucine is achieved, providing more protein/leucine does not further increase MPS [144], a phenomenon known as the “muscle-full effect” [145].

An important limitation to highlight before applying the data regarding the amount of protein required per meal to stimulate MPS in the elderly optimally is that all the included studies in that analysis involved the consumption of isolated high-quality protein sources [138]. This deserves attention because, in a “real-world setting,” proteins are often consumed in the context of mixed meals, which in turn may influence the resultant postprandial aminoacidemia and leucinemia. This scenario could influence the MPS response since the addition of other macronutrients, and the meals’ solid consistency, may reduce the aminoacidemia/leucinemia [146] and slow down its increase [147], possibly reducing MPS [148,149,150,151]. This thesis is supported by a recent trial that compared the ingestion of 35 g and 70 g of protein consumed in the context of mixed meals [152]. In this case, despite previous analysis showing near maximal stimulation of MPS in the older after the ingestion of 30–40 g of high-quality isolated protein, the authors reported more significant increases in MPS after the ingestion of 70 g compared with 35 g of protein. Therefore, considering the confidence interval of 0.21–0.59 g/kg/meal presented by Moore et al. [138], we recommend that in the context of mixed meals, older people should aim to consume the higher end, meaning at least 0.6 g/kg/meal of protein.

Despite our recommendations of a higher protein intake per mixed meal in the elderly (at least 0.6 g/kg/meal), we acknowledge that it may become difficult to achieve since the elderly may suffer from poor dentition, loss of appetite, and a condition called “anorexia of aging” [153]. Thus, new strategies have been studied over the last few years to overcome these barriers. For example, researchers started to test if fortification of suboptimal doses of protein with free leucine would induce similar increases in MPS compared with protein doses than previously evidenced near maximal stimulation of MPS [146,154,155,156,157,158,159,160,161]. One of the first studies that sought to evaluate the effects of leucine fortification on suboptimal doses of protein in older adults was conducted by Wall et al. (2013) [154]. Healthy older men were fed with 20 g of casein protein added or not with 2.5 g of crystalline leucine to evaluate the MPS response for the next 4 h. As a result, despite both groups consuming the same amount of protein, the group that received free leucine increased MPS by 22% more than the protein-only group. Later, Murphy et al. (2016) [146] investigated the integrative response of MPS by adding 5 g of leucine or placebo in each main meal consuming either 0.8 or 1.2 g/kg/d for six consecutive days. In addition, unilateral resistance exercise was also performed to understand the impact of leucine alone and leucine plus resistance exercise. No differences in MPS were observed between consuming high or low amounts of protein. However, the leucine-supplemented group showed higher rates of MPS both at rest and after resistance exercise. Importantly, resistance-exercised legs, accompanied or not by added leucine, showed higher rates of MPS when compared with rested legs, even in the supplemented condition [146].

This study highlights that resistance exercise is the most potent anabolic stimulus capable of raising MPS rates and should be the primary focus in interventions focused on maintaining or increasing skeletal muscle mass and function. Finally, it is noteworthy that both diets showed the same integrative MPS despite a difference of 50% in total daily protein intake between groups. It is possible that even 1.2 g/kg/d may not be enough to adequately stimulate MPS throughout the day since other research showed higher rates of daily MPS when consuming 1.5 g/kg/d, compared with the current RDA [161].

Long-term studies were also conducted to investigate the chronic effects of leucine co-ingestion with suboptimal protein doses on muscle mass and function measures. The first examined the influence of adding 2.5 g of leucine three times per day with the main meals in healthy older men for three months [162]. As a result, no effects were found on lean body mass. Posteriorly, the same group conducted a similar trial with the same supplementation protocol in type 2 diabetic older men and found no effects on muscle mass and function after six months [163]. More recently, Murphy et al. [159] recruited 107 men and women aged >65 to receive, two times daily, a supplement containing 10 g protein and 3 g total leucine (LEU-PRO); 10 g protein, 3 g total leucine, and 2 g ômega-3 (LEU-PRO + n − 3); or isoenergetic control (CON). Appendicular lean mass, strength, muscle function, and integrated rates of myofibrillar MPS were measured. After 24 weeks, the authors did not observe any effect of the supplementation protocol on the variables analyzed. Similar results were found when frail and pre-frail older individuals ingested 2.5 g of leucine with breakfast and lunch/dinner, combined with RT two times per week for 16 weeks [160].

Although it may seem contradictory to the acute and short-term data previously presented, some design limitations may have influenced the negative results. First, 3–6 months may be insufficient to detect a measurable difference between groups regarding the maintenance of muscle mass. Second, previous data have shown that a meal containing only 3 g of leucine could not raise integrated myofibrillar MPS at rest [157]. Thus, although contradictory to acute and short-term data, the amount of leucine-supplemented in these chronic studies may have been insufficient to induce an efficient anabolic stim-ulus.

Another important aspect regarding leucine supplementation is that, although it may trigger the anabolic signaling and is a crucial element of stimulating MPS, supplementing leucine alone seems ineffective to promote a positive muscle protein balance [164]. A fascinating study conducted by Van Vliet and colleagues [164] compared the effects of protein and isolated leucine supplementation on muscle protein turnover in 28 middle-aged women during a hyperinsulinaemic-euglycaemic clamp procedure (HECP). The supplemented groups consumed either 20 g of whey protein (containing 2.4 g of leucine) or 2.5 g of isolated leucine. As a result, only the whey-supplemented condition was able to increase MPS significantly. On the other hand, MPB decreased by 20% in both groups, a consequence of the HECP, since insulin has an antiproteolytic action [165].

Consequently, the resultant muscle protein balance of whey and isolated leucine-supplemented conditions was positive and neutral, respectively. The most probable explanation for the lack of stimulus of isolated leucine on MPS is that all EAA are necessary to provide the “building blocks” to the new proteins that will be synthesized [134]. Thus, although leucine may start the anabolic process, all EAA seem necessary to sustain it over time.

Given the available data in the literature, there is a need for new long-term studies evaluating higher doses of isolated leucine, combined with suboptimal amounts of protein in the context of a mixed meal to confirm the data presented in acute and short-term studies showing beneficial effects of isolated leucine on MPS. For now, despite the lack of studies showing benefits of chronic leucine supplementation, as an alternative to older people unable or unwilling to consume a minimum of 0.6 g/kg/meal of protein, we suggest that each main meal should contain, at least, 5 g of leucine, which can be reached via isolated leucine supplements.

### 5.2. Protein Quality

Quality is an important aspect that needs to be discussed when discussing protein intake. While protein quality does not seem to be determinant to younger individuals consuming an adequate amount of protein daily [166], it is possible that in older people, the quality of the protein source might have a more significant impact on anabolic responses and maintenance of muscle mass since anabolic resistance of aging decreases the sensitivity of older muscle to protein/amino acids ingestion.

Besides protein quantity, factors associated with protein quality, such as digestion and absorption kinetic (i.e., bioavailability) and amino acid composition (especially EAA content), directly interfere with its ability to stimulate the anabolic process and elevate the rates of MPS [167]. These characteristics of each protein source will dictate the postprandial rise in EAA concentrations (namely, amino-acidemia) and leucine concentrations (namely leucinemia), which are of utmost importance to increase MPS [146]. These components will affect protein digestion rate, how much of proteins’ amino acid content is effectively absorbed and directed to peripheral tissues (bioavailability), and the ability of this protein to stimulate anabolic signaling cascade [167]. All these steps combined will dictate the postprandial rise in amino-acidemia/leucinemia and MPS [167]

Due to anabolic resistance and knowing the critical role of EAA, particularly leucine, it would be important to guarantee enough of this amino acid to overcome this condition properly. In this sense, different proteins have been studied over the years to understand the anabolic properties of each source. Plant-based proteins are often classified as having lower digestibility and poorer EAA profiles [168]. Therefore, they may not be optimal to support skeletal muscle anabolism, particularly in the elderly.

Regarding absorption kinetics, the speed of amino acid absorption seems to directly influence the postprandial MPS, contributing to the development of the fast and slow protein model [169]. Despite studies showing superior effects of fast versus slow protein digestion and absorption rates on MPS [148,151,170], it should be highlighted that the protein sources consumed in those studies were isolated protein supplements. For example, Burd et al. (2012) [170] compared the ingestion of whey versus casein, both at rest and after resistance exercise, and showed a higher anabolic response in whey conditions in both situations. Pennings et al. (2011) [148] also tested whey against casein and hydrolyzed casein in older adults. In agreement with the results from Burd et al. (2012) [170], they found more outstanding MPS stimulation after whey ingestion compared to both types of casein protein. Notably, the higher concentration of plasma leucine combined with peak leucinemia resulting from whey supplementation showed a strong relationship with MPS, which corroborates the thesis of the fast and slow protein model. These data provide a better understanding of protein metabolism; however, they may be of limited applicability in a real-life context. In a real-world setting, proteins are often consumed together with other macronutrients and in the form of whole foods and mixed meals, which implies digestion and absorption rates similar to a slow protein supplement (i.e., casein) [147]. Therefore, fast digestive proteins should be prioritized when consuming protein supplements. At the same time, mixed meals tend to resemble slow digestive protein supplements, which could imply that higher amounts of protein would be required to stimulate MPS optimally.

### 5.3. Protein Source

Protein’s quality is as important as its source. There is an increasing interest in evaluating the differences between the anabolic potential of different protein sources. To date, most studies have been conducted with animal-derived protein and, in general, have shown better anabolic stimulus than plant protein on a gram-for-gram basis [167]. Yang et al. compared the effects of graded doses of isolated whey and soy protein in older individuals at rest and after resistance exercise [136]. To this end, supplements containing 0, 20, or 40 g of protein were ingested. The results showed that, in both situations, whey protein was significantly more efficient in increasing the fractional synthetic rate (FSR) than soy protein. On the other hand, soy protein could not increase FSR in any dose at rest. In addition, even after resistance exercise, 40 g of soy protein was less effective than 20 g of whey in inducing muscle anabolism. The lower capacity of soy protein than whey in stimulating MPS may result from lower leucine/EAA content, given that both proteins are considered fast digestive proteins. Thus, it could be argued that increasing the amount of total protein/leucine content to match whey’s content could be an alternative to overcome the lesser anabolic potential. However, to date, we are unaware of any study conducted in older adults that fortified suboptimal doses of soy protein with free leucine and measured MPS.

Supporting this idea, Gorissen et al. (2016) [171] analyzed the effects of wheat protein compared with casein and whey protein, in older men. The subjects ingested 35 g of whey, casein, or wheat protein, with leucine content of 4.4, 3.2, and 2.5 g, respectively. The results showed that despite the postprandial rise in leucine being greater after whey than other proteins, only casein stimulated MPS. Interestingly, casein and wheat protein resulted in similar peak leucinemia. However, casein ingestion induced more sustained leucinemia, which may have been responsible for MPS stimulation.

Nonetheless, increasing the amount of wheat protein to 60 g to match the leucine content of whey also prolonged the rise in leucine concentrations and stimulated MPS to levels like those of casein. These data are important because they demonstrate that peak aminoacidemia/leucinemia is essential, and the duration of the postprandial rise in amino acids plays a role in determining the anabolic properties of a protein source. Moreover, it suggests that increasing the amount of leucine in a lower-quality protein source may compensate for its reduced leucine content and seems to modulate aminoacidemia. It also suggests that increasing the amount of leucine in a meal may be required to increase the peak and/or sustain plasma leucine concentrations for more extended periods Gorissen et al. (2016) [171]. Both scenarios seem necessary to stimulate postprandial MPS properly.

Regarding the debate about different dietary protein sources and skeletal muscle anabolism, a recent systematic review and meta-analysis conducted by Morgan and colleagues [172] sought to determine the effects of protein source/quality on acute MPS and changes in lean body mass and strength, combined or not with resistance exercise in both young and older adults. The authors analyzed the impact of protein feeding alone on MPS, protein feeding combined with a bout of resistance exercise on MPS, and protein feeding combined with longer-term RT on lean body mass and strength. The comparisons were made with dose-matched proteins. Notably, most of them were made utilizing isolated protein sources, with doses ranging from 15–40 g of protein and 1.8–4.4 g of leucine. The results showed that protein quality significantly impacted MPS at rest only in the elderly [133]. On the other hand, when combined with resistance exercise, protein quality significantly impacts MPS in both young and older individuals. Regarding body composition and lean body mass, despite no influence of protein quality being found, only three studies included that investigated older adults.

More recently, Roschel et al. (2021) [160], in a double-blind, randomized, placebo-controlled trial, investigated the effects of whey, soy, or isolated leucine combined with RT in measures of physical function, lean mass, and cross-sectional muscle area of pre-frail and frail elderly. After 16 weeks, neither supplemented group showed more significant improvement in any parameter compared with the placebo. One could argue that some limitations may have influenced the negative results. For example, total daily protein intake by protein-supplemented groups may have remained too low to provide additional benefits (1.2 g/kg/d); isolated leucine was provided in doses lower than those that have shown marked increases in integrated MPS [157,173]; and protein dose per meal may have been below the threshold to stimulate MPS, especially in the soy supplemented group.

Therefore, given the apparent higher acute anabolic potential of higher-quality protein sources on MPS and a limited number of chronic studies in older adults, more long-term investigations need to be conducted to better understand the impact of different protein sources/quality on the maintenance of skeletal muscle mass over time.

Besides the proposed lesser digestibility and amino acid absorption kinetics, amino acid composition plays a determinant role regarding plant protein’s lower acute MPS response when directly compared with a matched dose of animal protein in the elderly [167]. Pinckaers et al. (2021) recently reported a comprehensive analysis of different animal and plant-based proteins. They revealed that the EAA content of plant-based proteins is generally lower when compared with animal-derived protein, besides being often deficient in one or more specific EAA, such as leucine, lysine, and/or methionine [167]. This poorer EAA profile theoretically limits the postprandial rise in MPSc [134].

Several nutritional strategies have been proposed to overcome the lesser anabolic potential of plant-based proteins. In short, foods with lower protein content can be optimized through extraction and processing protein to produce isolated proteins in lower amounts of food portions. Another alternative is fortifying proteins with the specific limiting amino acid. Protein blends can also be formed by combining proteins that are complementary in their limiting amino acids. Finally, another strategy is increasing the portion of total protein intake to reach the required EAA amount. However, this could be challenging for older individuals depending on the protein source [167].

Another important gap in the literature is the lack of studies evaluating anabolic response to whole foods, mainly plant-based proteins. It is well-known that the food matrix and other nutrients present in whole food may influence the anabolic response [174,175]. Moreover, caloric density and other macronutrients also influence postprandial aminoacidemia [176], which can also influence MPS. Thus, future studies should expand on understanding the anabolic properties of whole foods and mixed meals to provide better recommendations based on more “real-life conditions” studies.

New protein sources have been studied as alternatives to animal proteins in recent years. Studies evaluating the anabolic response to a fungi protein, namely, “mycoprotein”, were conducted in both young [177] and older individuals [178]. Mycoprotein has high protein density with similar amino acid content to dairy proteins and seems to present good digestibility [179]. Monteyne and colleagues (2020) determined whether a non-animal-derived diet can support daily myofibrillar MPS to the same extent as an omnivorous diet [178]. In this case, 19 older adults were allocated to consume 1.8 g/kg/d of either 71% of protein predominantly from animal sources or exclusively vegan protein sources (57% from mycoprotein). They also performed daily unilateral resistance exercise. The results showed comparable daily free-living rates of muscular MPS between omnivorous and vegan diets in both rested and exercised conditions. These results suggest that a vegan diet composed predominantly of mycoprotein is similarly effective to an omnivorous diet to support skeletal muscle anabolism in the elderly [178]. However, it is noteworthy that subjects consumed 1.8 g/kg/d of protein; thus, we do not know the MPS response to lower total daily protein intake; a relevant issue since consuming 1.8 g/kd/d of protein may not always be feasible. Nevertheless, these data corroborate other acute studies, suggesting that consuming higher amounts of non-animal-derived protein sources may compensate for their lower-quality score.

Another protein source that has received much attention in the last few years is collagen. Collagen products are often proposed to be especially useful for healthy aging, and many supplements and medical drinks in the market, specific for the elderly, contain collagen. However, data regarding its effectiveness for healthy aging do not seem to corroborate the industry’s marketing. Oikawa et al. (2020) [180] investigated the effects of whey and collagen protein, combined with unilateral resistance exercise, on acute and long-term MPS in 22 healthy older women [180]. To this end, subjects were randomly assigned to consume 30 g of either whey or collagen protein twice daily for six days to determine acute and integrated MPS responses. The results showed that whey protein increased acute MPS more than collagen protein both at rest and after resistance exercise. In addition, measures of integrated MPS revealed that only whey protein was able to significantly elevate daily MPS, whereas no effect was observed with the ingestion of collagen.

These findings corroborate data from a previous study conducted by the same research group that showed higher rates of integrated MPS and better recovery of leg lean mass with consumption of whey versus collagen protein in healthy older individuals after a period of energy restriction and step reduction (<750 daily steps) [181]. The results of these two studies regarding the anabolic potential of whey and collagen proteins might be explained by the differences in the quality of the protein sources consumed. While whey is a high-quality, fast digestive protein [29], collagen has very low amounts of leucine and lacks tryptophan. Therefore, it is considered an incomplete protein since it lacks at least one EAA [29].

In contrast with those findings, a recent long-term randomized controlled trial [129] evaluating the effects of carbohydrate, whey, and collagen supplementation on muscle size and strength of healthy older individuals did not find any differences between any condition after one year of supplementation. In this trial, groups consumed two daily doses of carbohydrate or protein supplements containing 20 g, which increased consumption of 27 and 32 g of protein per day in the whey and collagen groups, respectively. As a result, total daily protein intake increased from 1.1 to 1.5 g/kg/d; however, there were no benefits in maintaining muscle mass and strength. These data contradict observational, acute, and short-term studies that suggest that an increase in protein intake alone does not benefit in preserving muscle mass or strength in healthy older adults. Additionally, a question has been raised regarding the role of protein quality since acute studies demonstrate the superior anabolic potential of whey versus collagen. However, the present investigation does not show any advantage of higher-quality protein regarding long-term muscle health. More long-term studies evaluating higher versus lower protein diets with varying protein sources/quality need to be conducted to better understand the long-term impacts of such different proteins and provide better practical recommendations [129]. Given the available literature, we recommend ingesting higher-quality proteins for acute and short-term studies that clearly showed a superior anabolic response than lower-quality protein. Unfortunately, there is a lack of long-term investigations despite the incongruent results between short- and longer-term studies. A summary of the different types of protein sources commercially available can be found on Table 1.

### 5.4. Protein Distribution

Currently, there is an increasing debate regarding the role of protein consumption patterns throughout the day and the optimization of skeletal muscle mass [182]. Specifically, some authors propose that a more even protein consumption pattern would be a better option to optimize muscle anabolism when compared with a more skewed protein intake [146]. This concept emerged from studies evaluating the acute and short-term responses of MPS to different patterns of protein intake throughout the day [183,184]. For example, after a protein-containing meal, there is an increase in postprandial MPS that persists up to 3–5 h, depending on the ingested protein source [185]. It is also known that the skeletal muscle capability to utilize dietary amino acids to stimulate MPS after protein ingestion presents a saturable dose. Thus, consuming protein beyond its capacity will not result in an additional MPS increase [138], a phenomenon known as the “muscle full effect” [145].

Given this scenario, it was hypothesized that consuming dietary proteins in sufficient amounts to stimulate MPS in each main meal would be more efficient than consuming higher amounts in just one or two meals during the day. In other words, an even protein intake distribution in each main meal containing enough protein to stimulate MPS properly would be more beneficial than consuming a more skewed pattern. The latter would result in at least one or two meals with suboptimal protein amounts that would not be enough to provide an efficient anabolic stimulus [145].

The first study to directly test this hypothesis was conducted by Areta et al. (2013) [183], in which different protein ingestion patterns were compared over a 12 h period in young adults. MPS was evaluated after consuming 80 g of whey protein in three different protocols: 8 × 10 g every 1.5 h, 4 × 20 g every three hours, or 2 × 40 g every six hours. The results showed that the intermediate pattern was better, which is in accordance with the saturable dose of protein in young adults consuming isolated protein sources and the “muscle-full effect” theory) [183].

Additionally, Mamerow et al. (2014) [184] examined the effects of protein distribution on 24-h MPS in healthy adult men and women. The authors measured the MPS changes in response to isoenergetic and isonitrogenous diets, with the amount of protein evenly distributed (~30 g each meal) throughout the main meals or skewed (10, 16, and 63 g at breakfast, lunch, and dinner, respectively). Analysis revealed that the 24-h mixed MPS rate was 25% higher in the even distribution compared to the skewed pattern [183,184]. The recent publication of chronic data evaluating this apparent superiority of a more balanced protein distribution over muscle protein anabolism was published by Yasuda et al. (2020) [186]. In a 12-week parallel group, RCT, 26 men were assigned to perform three sessions of RT per weak and consume either an evenly protein distribution pattern (0.33, 0.46, and 0.48 g/kg in each main meal) or a more skewed distribution (0.12, 0.45, and 0.83 g/kg in each main meal). In both conditions, the total daily protein intake was around 1.3–1.4 g/kg/d [186]. The results showed higher increases in lean mass in those consuming a more balanced protein pattern, which is observed in both acute and short-term studies. However, it is important to highlight that neither group consumed the recommended daily protein intake to optimize skeletal muscle hypertrophy, a minimum of 1.6 g/kg/d of protein [10]. Therefore, we do not know if there will be any effect of protein consumption pattern once total daily protein intake recommendations are met. Collectively, the available acute and short-term studies, at least in younger individuals, support the hypothesis that a more balanced distribution of protein intake throughout the day seems to be beneficial to optimize skeletal muscle mass.

Regarding the elderly population, some observational data have been published evaluating the influence of the protein ingestion pattern on skeletal muscle mass maintenance. Loenneke et al. (2016) [187] found that consuming 30–45 g of protein per meal was positively associated with greater leg lean mass and knee extensor muscle strength. Bollwein et al. (2013) [188] reported that in a sample of non-frail, pre-frail, and frail older subjects, non-frail individuals exhibited a more evenly distributed protein consumption pattern, despite the similar total daily protein intake in all three groups [188]. More recently, Hayashi et al. (2020) [189] found that the number of meals containing either >20 g or >30 g of protein was significantly associated with greater total and appendicular lean mass. However, Hudson et al. 2020 [182] conducted a comprehensive literature review regarding the impacts of protein consumption patterns on long-term muscle mass maintenance in the elderly. They highlighted some inconsistent findings regarding its impact. The authors argued that when total protein intakes are above the RDA (0.8 g/kg/d), a more even protein consumption may be superior to a skewed pattern to better support skeletal muscle health. In the meantime, when consuming less than the RDA, an unbalanced protein intake may be more advantageous since at least one meal would contain enough protein to reach the leucine threshold to stimulate MPS properly. However, they also argue that instructing older individuals to consume a more balanced protein intake may be a strategy to increase their total daily protein intake to levels higher than 0.8 g/kg/d.

Acute studies have also been conducted in older individuals to evaluate the impact of different protein intake patterns on muscle protein metabolism. For example, Kim et al. (2015) [161] examine MPS following protein ingestion in mixed meals at two doses of protein and two intake patterns. They randomly assigned 20 healthy older subjects to one of the following conditions: protein intake of 0.8 g/kg/d in an even or uneven condition (1RDA-E, 1RDA-U, respectively) and protein intake of 1.5 g/kg/d in an even or uneven condition (2RDA-E, 2RDA-U, respectively). Regardless of distribution patterns, the results showed greater MPS with 2RDA versus 1RDA. This finding contradicts the acute studies in younger individuals. However, some issues need to be highlighted. First, the study sample was very small, with only 4–6 participants per group. Second, the 2RDA group consumed around 0.5 g/kg/meal of protein, which may be insufficient to properly stimulate MPS in older individuals in the context of mixed meals. The data provided by Moore et al. (2015) [138] suggested that 0.4 g/kg/meal of high-quality isolated protein would be sufficient to optimally stimulate MPS in older adults. However, in a real-world setting, protein is usually consumed as whole foods and in the context of mixed meals. As previously discussed, these meals present higher caloric density and different degrees of protein quality, both factors that directly influence postprandial amino-acidemia/leucinemia and may compromise anabolic response [176,183,185]. Thus, consuming 0.5 g/kg/meal may be suboptimal to provide an efficient anabolic stimulus. Therefore, older adults would probably have to consume, at least, the higher end of confidence interval provided by Moore et al. (2015) [138], which is 0.6 g/kg/meal.

Later, the same research group conducted a chronic yet short-term study to examine the effects of an even versus skewed protein consumption pattern on body composition and protein metabolism [190]. During eight weeks, elderly individuals consumed 1.1 g/kg/d of protein in a balance or skewed pattern considering the three main meals. The results revealed no impact of consumption patterns on MPS and body composition. As discussed above, the amount of protein in each meal of the even distribution group contained around 0.37 g/kg, which is insufficient to optimally stimulate MPS considering high-quality isolated proteins. In this case, subjects consumed whole foods in mixed meals, which may increase the per-meal protein requirement to maximize MPS. Thus, the data from these two studies suggest that a consistent distribution pattern does not necessarily result in optimized muscle anabolism.

A higher total daily protein intake is necessary to allow a balance protein intake to be theoretically optimal. It also seems to have been the case for the study conducted by Buckinx et al. (2019) [191], in which 30 sedentary and obese older men and women were divided into two groups. In the first (P20), the subjects consumed <20 g of protein in at least one meal. In the second (P20+), the subjects consumed >20 g of protein every meal. The intervention also included a high-intensity interval training (HIIT) program three times per week and lasted a total of 12 weeks. The results showed that both groups reduced waist and hip circumferences and improved functional capacity without any effect from the protein consumption pattern. This result is not surprising since the subjects in the P20+ group had a mean per meal protein consumption of 20–25 g, which is below the minimum the 0.4 g/kg/d recommended by Moore et al. (2015) [138].

Therefore, to date, no chronic study has shown benefits from consuming a balanced protein intake throughout the day. However, it is likely that no studies have tested protein doses that would effectively stimulate MPS under conditions of whole foods and mixed meals [138]. Future studies need to investigate integrative MPS of balanced versus skewed protein consumption in mixed meals with higher protein doses per meal. Additionally, chronic studies with at least six months of consuming higher protein amounts per meal (at least 0.6 g/kg/meal) are required to confirm whether there is a benefit of a more balanced consumption pattern and in what magnitude. Moreover, other alternatives to high protein intake need to be examined. Long-term studies evaluating higher doses of isolated leucine supplementation to main meals are also required to provide more feasible alternatives to the elderly that are unable or unwilling to consume such high amounts of protein per day/per meal.

### 5.5. Protein Intake and Resistance Training (RT)

Aside from pharmacological therapy, resistance exercise provides the most efficient anabolic stimulus to skeletal muscle tissue growth [10]. It is known that a bout of resistance exercise stimulates both MPS and MPB [192]; however, in the absence of nutritional support (i.e., EAA), muscle protein balance remains negative [192]. On the other hand, combining resistance exercise with protein/EAA ingestion results in a positive muscle protein balance since rates of MPS exceed MPB [193]. Like the rest of the conditions, there is a dose-response regarding the effects of protein intake after resistance exercise on postprandial stimulation of MPS. Yang and colleagues [9] recruited 37 older men to complete a bout of unilateral leg-based resistance exercise before ingesting 0, 10, 20, or 40 g of whey protein and analyzed postprandial MPS after four hours. After a resistance exercise session, they found that 40 g of whey intake increased myofibrillar MPS by 91%, while 20 g of whey exhibited only a 44% increase above basal levels. Posteriorly, Holwerda et al. (2019) [194] assessed the postprandial MPS response to the ingestion of placebo, 15 g, 30 g, and 45 g of milk protein concentrate after a bout of resistance exercise in 48 healthy older men. As a result, incorporating dietary protein-derived amino acids into de novo myofibrillar MPS showed a dose-dependent increase after ingesting graded doses of milk protein concentrate [194]. Findings from Pennings et al. (2011) [195] add to these data, showing that performing exercise before protein ingestion allows for greater use of dietary amino acids for de novo MPS in older adults [195].

Although acute data clearly show a synergistic effect of resistance exercise and protein intake on anabolic stimulation of skeletal muscle protein metabolism, chronic investigations combining resistance exercise and increased protein consumption have yielded mixed results [194,195]. In a randomized, double-blind, placebo-controlled trial, 62 frail elderly subjects participated twice weekly in a progressive resistance exercise program. They were supplemented twice daily with either milk protein (2 × 15 g) or placebo [125]. The supplemented group increased their total daily protein intake from 1.0 to 1.3 g/kg/d. As a result, lean body mass increased 1.3 kg in the protein-supplemented group, while no change was observed in the placebo group. A meta-analysis also corroborates this finding by showing significantly, although modestly, higher lean mass gains than controls [196,197]. However, this notion was recently questioned with a publication of a new meta-analysis by Morton et al. (2018) [10], which suggests that the additional effects of increased protein consumption on lean mass gains seems to be reduced, and even eliminated, with advancing age. Nevertheless, some issues regarding the included studies must be highlighted. First, the average supplemental daily protein dose given to older participants was only 20 g, far below the recommended dose of 0.4 g/kg/meal of a high-quality isolated protein due to anabolic resistance [160]. Additionally, there were a low number of studies conducted in older adults; a lack of complete information was reported by those studies regarding baseline protein intakes, and finally, the daily protein ingestion was below the suggested level for older individuals (1.2–1.6 g/kg/d). Low doses of post-exercise supplemental protein may have influenced the lack of benefits of increased protein consumption on changes in lean mass in the elderly.

Two recently published randomized controlled trials also did not show any effects of supplementing protein combined with resistance exercise to improve lean body mass gains [129,160]. In a 16-week study, Roschel et al. (2021) [160] supplemented frail and pre-frail elderly individuals with two daily doses of 15 g of placebo, whey, and soy protein, consumed together with breakfast and dinner. Markers of physical function, body composition, and qCSA were measured and did not differ between the groups. Total daily protein consumption rose from 0.8 to 1.2 g/kg/d [160]. As previously discussed, this value does not seem to be enough to increase muscle anabolism optimally.

The other investigation was, to our knowledge, the longest trial to date to evaluate the effects of increased protein consumption on measures of muscle mass and function [129]. Participants were enrolled and randomized into one of the following five groups: (1) carbohydrate supplementation: two daily doses of 20 g maltodextrin; (2) two daily doses of 20 g of whey protein; (3) collagen supplementation: two daily doses of 20 g of collagen protein; (4) light-intensity training with the same protocol of whey protein supplementation; and (5) heavy RT with the same protocol of whey protein supplementation. Participants should ingest their supplements in the morning and midday together with meals. The results showed only the groups that exercised increased qCSA, strength, and power. In addition, heavy RT was the only group that slightly increased lean body mass [129]. Unfortunately, this study lacked a group to evaluate only RT effects; thus, we cannot isolate the effects of protein intake and RT. However, given that the whey-supplemented-only group did not exhibit improvements in any parameter, and both light and heavy exercise showed benefits in variables analyzed, one could argue that exercise was responsible for these improvements.

In conclusion, the inconsistencies regarding the role of increased protein consumption to potentiate resistance exercise effects on skeletal muscle mass and function may be related to differences in study design, such as total daily protein intake, per-meal protein dose, protein quality, and population studied (healthy, pre-frail, and frail older individuals). More long-term studies combining resistance exercise with increased protein consumption must be conducted. There is only one trial with a one-year duration to this date. It is fundamental to highlight that resistance exercise should be the primary intervention to increase skeletal muscle mass and function in older adults. Increased protein intake seems to play a supporting role in maintaining skeletal muscle health.

## 6. Practical Recommendations

Given the amount of research available and some conflicting information, we summarize here and in Figure 1 some practical recommendations for optimal protein intake and muscle contraction in the aging population:(1)Total daily protein intake around 1.6–1.8 g/kg/d;(2)Three main meals containing 0.6 g/kg of high-quality protein sources;(3)At least 5 g of leucine per meal;(4)When protein supplementation is necessary, prioritize high-quality, fast digestive protein (i.e., whey);(5)Despite acute and short-term evidence showing benefits of isolated leucine supplementation to mixed meals, more long-term data are required to recommend supplemental leucine properly;(6)Ensure adequate energy supply to avoid negative energy balance since it reduces post-prandial MPS to protein ingestion and exacerbates anabolic resistance;(7)Resistance exercise at least twice a week;(8)Reduce sedentary time.

## 7. Conclusions

This review highlighted different aspects of optimizing nutritional strategies focused on maintaining skeletal muscle mass in the aging process. Specifically, we discussed how one could theoretically maximize the anabolic potential of protein meals by managing protein dose, protein quality, and pattern of consumption throughout the day. To this end, based on available data in the literature, it seems reasonable to recommend older people to consume 1.6–1.8 g/kg/d of protein, and attention should be paid to total daily protein intake. However, observational and short-term studies point to benefits in consuming at least 0.6 g/kg/meal of protein evenly distributed in three main meals containing 5–6 g of leucine, minimum.

We acknowledge that it can be challenging for older adults to consume such high amounts of daily protein. Thus, we reiterate the need for studies evaluating other feasible nutritional strategies for older people to adhere to. In this sense, more long-term isolated leucine supplementation research evaluating doses of at least 5 g per meal is mandatory. Protein supplements may be a viable option to increase total daily and per-meal protein intake since a recent systematic review and meta-analysis revealed no significant impact on appetite, and reductions in energy intake, in the elderly [198]. In addition, acute and short-term studies also point to isolated leucine as a good alternative to provide an efficient anabolic stimulus. Thus, we reinforce the need for studies evaluating long-term effects of higher doses of isolated leucine, ingested with main meals, to evaluate its effectiveness to maintain muscle mass.

Another fundamental aspect that needs to be highlighted is the maintenance of energy balance. Several studies have shown that negative energy balance decreases MPS response to protein feeding, exacerbating anabolic resistance [199,200,201]. Thus, ensuring adequate energy intake in older adults is fundamental to preserving skeletal muscle mass over time.

Although beyond the scope of this review, it is crucial to highlight studies investigating the effects of combined different nutrients and supplements in MPS concentrations and muscle retention in older individuals. Many studies [159,202,203,204,205,206,207], including a recently published meta-analysis [208], have combined protein, leucine, vitamin D, ômega-3, and creatine associated or not with RT. Regarding its effectiveness, vitamin D consumption seems to be beneficial only when there is an insufficiency or deficiency, with no further benefits when levels are higher than those recommended [209]. On the other hand, ômega-3 supplementation studies have produced controversial results, and further work is needed to provide more robust recommendations [209]. Lastly, creatine is one of the most studied supplements, consistently showing positive improvements in strength, physical function, and lean body mass when combined with RT in older adults [210]. Although it became impossible to isolate the effects of the individual nutrients, the idea behind the consumption of the “anabolic cocktail” is to overcome the heterogeneity of individual responses to each nutrient. Since each individual will respond differently to each nutrient, being more or less responsive to its anabolic activity, combining all may overcome these differences and provide the most efficient anabolic stimulus. Therefore, combining different nutrients with protein supplements may be a feasible and practical way to improve skeletal muscle mass and function in older people.

## Figures and Tables

**Figure 1 nutrients-14-00052-f001:**
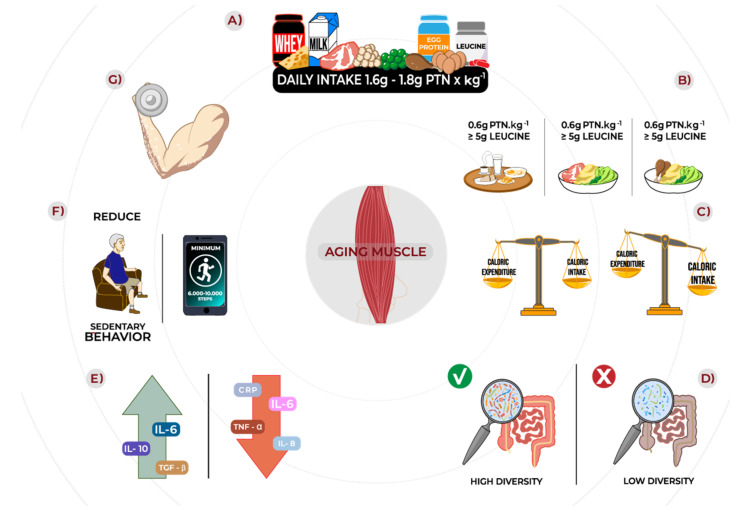
Maintenance of skeletal muscle health in aging. (**A**) Daily protein consumption of 1.6–1.8 g·kg^−1^ body weight, prioritizing high biological value protein through food or supplements. (**B**) Consumption of three daily protein meals containing 0.6 g of PTN·kg^−1^ or at least 5–6 g of leucine. (**C**) Positive or neutral daily energy balance. (**D**) Maintenance of intestinal diversity. (**E**) Reduced or controlled pro-inflammatory state-higher concentration of anti-inflammatory markers (muscular IL-6, IL-10, and TGF—β) compared to pro-inflammatory factors (adipocyte IL-6, IL-8, CRP, and TNFα). (**F**) Reduced sedentary behavior (≤6 h·day^−1^) or 6000–10,000 steps daily. (**G**) Resistance training (RT) at least twice a week.

**Table 1 nutrients-14-00052-t001:** Characteristics of different commercially available protein sources.

Protein Source	EAAs Profile	Leucine Content	Digestion Rate	Bioavailability
Whey	Complete	High	Fast	High
Casein	Complete	High	Slow	High
Milk	Complete	High	Slow	High
Isolated soy	Complete	Medium	Fast	Medium
Collagen	Incomplete	Low	Fast	Medium
Mycoprotein	Complete	High	Fast	High
Isolated wheat protein	Complete	Medium	Fast	High

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
