# Peer review of "Strategies to Prevent Sarcopenia in the Aging Process: Role of Protein Intake and Exercise"

_nutrients, 2021, doi:10.3390/nu14010052_

Round 1
Reviewer 1 Report
Comments to the review
“Strategies to prevent sarcopenia in the aging process: Role of protein intake and exercise”
The review paper is presented well, is detailed and covers most of the aspects of sarcopenia and its causative factors rooted in protein ingestion. However, the paper can greatly benefit from the following revisions:
1) While a review paper presents a predominantly textual review of the prevalent literature, the importance of figures and diagrams cannot be understated. The paper has only one diagram which makes it a cumbersome read. For instance, section 5.2. Protein Sources/Protein Quality can include a summary of these sources in tabular form.
2) Building on the previous point, it is also a good idea to summarize some of the important literature in a tabular form. This makes referring to literature easy for the reader and keeps information organized. Moreover, the sequence of the references should be organized in an increasing order, as usually is in the reviews/articles in literature.
3) Section 5.2. Line 755-774 should be removed as this information is not within the scope of the review.
4) The same section could also benefit from a re-sectioning and division into Protein Source and Protein Quality.
5) Conclusion and the recommendations offered can be split into two different sections for better structure
6) Building on the previous point, there is not enough clarity whether the recommendations are directed towards older adults or not.
Author Response
Dear Reviewer,
thank you for your contribution to our manuscript. Here is a point-by-point response to your comments. You can also see them in the new file uploaded attached.
1) A table was added to make the information clearer for the readers.
2)The literature is presented as they appear in the paper to make it easier for the readers to consult or obtain more details from the original paper.
3) After reviewing the highlighted section, we believe it is important to keep it in the text. We highlight leucine's acute effects and point out the importance of new studies focusing on chronic effects. The practical recommendation also takes into consideration what is exposed here. Therefore, we believe that removing this section from the text would impair its comprehension.
4) This section was reorganized for better understanding.
5) This section was reorganized for better understanding.
6) Yes, the entire review is focused on older adults. Parts were written again to make it clear.
Again, we appreciate your comments and contribution.
All the best,
Patricia S. Rogeri

Reviewer 2 Report
The authors have done a good job of reviewing the important and quickly advancing field of nutrition/exercise and their effects upon aging/sarcopenia. Furthermore, they have made well-reasoned recommendations for optimal aging. However, the manuscript requires substantial proof reading
Author Response
Dear reviewer,
thank you for your comments and contribution to our manuscript. An extensive review of the English language was done and you can view the new version of the manuscript attached.
All the best,
Patricia S. Rogeri
